

# Spatial patterns of aboveground phytogenic Si stocks in a grass-dominated catchment – Results from UAS based high resolution remote sensing

Marc Wehrhan[1], Daniel Puppe[2], Danuta Kaczorek[1], Michael Sommer[1,2,3]

[1] Leibniz Centre for Agricultural Landscape Research (ZALF), "Landscape Pedology" Working Group, 15374 Müncheberg, Germany
[2] Leibniz Centre for Agricultural Landscape Research (ZALF), "Silicon Biogeochemistry" Working Group, 15374 Müncheberg, Germany
[3] University of Potsdam, Institute of Geography and Environmental Science, 14476 Potsdam, Germany

*Correspondence to*: Marc Wehrhan (wehrhan@zalf.de)

**Abstract.** Various studies have been performed to quantify silicon (Si) stocks in plant biomass and related Si fluxes in terrestrial biogeosystems. Most of these studies were performed at relatively small plots with an intended low heterogeneity in soils and plant canopy composition, and results were extrapolated to larger spatial units up to global scale implicitly
assuming similar environmental conditions. However, the emergence of new technical features and increasing knowledge on details in Si cycling leads to a more complex picture at landscape or catchment scales. Dynamic and static soil properties change along the soil continuum and might influence not only the species composition of natural vegetation, but its biomass distribution and related Si stocks. Maximum Likelihood (ML) classification was applied to multispectral imagery captured by an Unmanned Aerial System (UAS) aiming the identification of land cover classes (LCC). Subsequently, the Normalized
Difference Vegetation Index (NDVI) and ground-based measurements of biomass were used to quantify aboveground Si stocks in two Si accumulating plants (*Calamagrostis epigejos* and *Phragmites australis*) in a heterogeneous catchment and related corresponding spatial patterns of these stocks to soil properties. We found aboveground Si stocks of *C. epigejos* and *P. australis* to be surprisingly high (maxima of Si stocks reach values up to 98 g Si m$^{-2}$), i.e., comparable to or markedly exceeding reported values for the Si storage in aboveground vegetation of various terrestrial ecosystems. We further found spatial patterns of plant
aboveground Si stocks to reflect spatial heterogeneities in soil properties. From our results we concluded that (i) aboveground biomass of plants seems to be the main factor of corresponding phytogenic Si stock quantities and (ii) a detection of biomass heterogeneities via UAS-based remote sensing represents a promising tool for the quantification of lifelike phytogenic Si pools at landscape scales.

# 1 Introduction

Biogenic silicon (BSi), i.e., silica precipitates (SiO$_2 \cdot n$H$_2$O) synthesized by various organisms, has been recognized as a key factor controlling Si fluxes from terrestrial to aquatic ecosystems (Dürr et al., 2011; Street-Perrott and Barker, 2008; Struyf



and Conley, 2012), which mainly results from its pool size and a larger solubility compared to silicate minerals (e.g., Cornelis and Delvaux, 2016). Eukaryotic and prokaryotic organisms, i.a., plants, bacteria, fungi, diatoms, testate amoebae, and sponges, are able to synthesize BSi (Clarke, 2003; Ehrlich et al., 2010), and corresponding BSi pools can be found in terrestrial

biogeosystems (Puppe et al., 2015; Puppe, 2020; Sommer et al., 2006). BSi structures of different origin indicate differences in their physicochemical surface properties (Puppe and Leue, 2018), which in turn control their dissolution kinetics (Bartoli, 1985; Fraysse et al., 2006, 2009). In most terrestrial ecosystems phytogenic Si, i.e., BSi synthesized by plants, generally represents the largest BSi pool in soil-plant systems, hence exerts the strongest influence on Si fluxes into soils.

For the majority of higher plants Si is considered as a beneficial element, because various positive effects of Si accumulation

in plants have been revealed, i.e., increased plant growth and resistance against abiotic and biotic stresses (e.g., Epstein, 2009; Ma and Yamaji, 2006; Puppe and Sommer, 2018). In this context, especially grasses of the family Poaceae (or Gramineae) are known as strong Si accumulators (Hodson et al., 2005), and corresponding Si storage in aboveground vegetation, e.g., in the Great Plains or the tropical humid grass savanna, has been found to be an important driver in Si cycling (Blecker et al., 2006; Alexandre et al., 2011). Various studies have been performed to quantify Si stocks and fluxes in/from the above- and

belowground plant biomass (e.g., Alexandre et al., 1997; Bartoli, 1983; Cornelis et al., 2010; Sommer et al., 2013; Turpault et al., 2018). Most of these studies were performed at (sequences of) small-scale plots ($<10^2$ m$^2$) with intended low heterogeneity in soils and plant canopy composition. Often results were extrapolated to larger spatial units up to global scale implicitly assuming similar environmental conditions (e.g., Carey and Fulweiler, 2012). However, the emergence of new technical features and increasing knowledge on details in Si cycling (e.g., the role of lateral fluxes) leads to a more complex picture at

landscape or catchment scales. Dynamic and static soil properties change along the soil continuum and might influence not only the species composition of natural vegetation, but its biomass distribution and related Si stocks.

Remote sensing represents an efficient tool to provide spatially consistent information on environmental objects, conditions and properties. To identify different land covers or to assess biodiversity indicators, supervised classification techniques such as Maximum Likelihood (ML) has found wide acceptance (Fuller et al., 1998; Otukei et al., 2010; Shafri et al., 2007; Strecha

et al., 2012; Gonzáles et al., 2015). A widespread method for the derivation of quantitative canopy properties is the use of vegetation indices (VIs) in combination with ground-based measurements (Thenkabail et al., 2002; Lelong et al., 2008; Zarco-Tajeda et al., 2012). VIs are linear, orthogonal or ratio combinations of reflectance calculated from different wavelengths ranging from the visible (VIS) to the near-infrared (NIR) part of the electromagnetic spectrum (Bouman, 1992) and found to be appropriate proxies for temporal and spatial variation in vegetation canopies and biophysical parameters (Gao et al., 2000;

Haboudane et al., 2004). In particular, the Normalized Difference Vegetation Index (NDVI), as the most commonly used VI, relates reflectance in red (sensitive to chlorophyll absorption) and near-infrared (sensitive to canopy and leaf structure) wavebands (Rouse, 1974). Numerous satellite based studies applied VIs to quantify biophysical vegetation parameters either of crops (Moran et al., 1995; Kross et al., 2015) grassland (Gammon et al., 1995; Wang et al., 2005) or pristine and near-natural ecosystems (Kim et al., 2015; Cui et al., 2018). The recent development of Unmanned Aerial Systems (UAS) offers

new options for high-resolution observations at landscape and catchment scale. Successful preprocessing workflows were





developed for UAS imagery as a prerequisite for accurate image interpretation (Laliberte et al., 2011; Berni et al., 2009; Kelcey and Lucieer, 2012; Lelong et al., 2008; Wehrhan et al., 2016).

UAS missions have been conducted over hardly accessible areas such as wetlands (Strecha et al., 2012; Zweig et al., 2015), riparian zones of lakes and rivers (Husson et al., 2014; Husson et al., 2016), estuarine tidal flats (Kaneko and Nohara, 2014)

and riparian forests (Dunford et al., 2009). Most of the studies delineated the patchy and small-scale distribution of plant communities and identified individual species by using of-the-shelf (partly modified) compact digital cameras providing an adequate sub-decimeter resolution in VIS and NIR spectral wavelengths. Zarco-Tajeda et al. (2012) demonstrated the successful application of a narrow-band multispectral sensor to assess water stress status of olive, peach and orange trees by estimates of chlorophyll fluorescence emissions. Turner et al. (2014) coupled multispectral and thermal imagery to investigate

the physiological state of Antarctic moss beds. Wehrhan et al. (2016) derived biomass patterns of lucerne (Medicago sativa) from UAS imagery in a soil landscape, which is strongly affected by soil erosion and Tóth (2018) observed seasonal and spatial changes of *Phragmites australis* derived from multi-temporal UAS imagery. Recently, Easterday et al. (2019) demonstrate the benefits of UAS-based remote sensing for plant water status estimates of shrubs using VIs derived from multispectral imagery. However, none of these studies addressed both the classification of species composition and the quantification of respective

aboveground biomass fractions. To the best of our knowledge there is no study published until now, which finally quantifies aboveground plant Si accumulation and its spatial distribution using UAS remote sensing in a heterogeneous catchment and relates the spatial patterns to relevant soil properties.

In the current study we apply UAS-based remote sensing to the grass-dominated, artificially catchment "Chicken Creek". It represents an ideal study site for Si cycling at catchment scale for several reasons: Firstly, the vegetation dynamics as well as

soil development have been intensively monitored ab initio (e.g., Elmer et al., 2013; Zaplata et al., 2011a, b). From this data base the site-specific appearance and spread of two predominant Si accumulators, *Calamagrostis epigejos* and *Phragmites australis* (both belonging to the Poaceae family), can be derived, which allows an estimation of mean annual Si uptake rates. Secondly, extensive soil data (repetitive sampling campaigns) are available at a 20 m x 20 m grid (Gerwin et al. 2011). Potentially important drivers for the observed spatial patterns of plants, like soil texture and nutrients can be withdrawn from

these data. Thirdly, previous studies already clarified the BSi pool dynamics in soils at Chicken Creek during initial pedogenesis (Puppe et al., 2014, 2016, 2017, 2018). Here we present a methodological approach to quantify the Si stocks of two Si accumulators (i.e., *C. epigejos* and *P. australis*) at catchment scale and their relationship to soil-related drivers. In detail we want to answer three major research questions:

(i) How large are aboveground phytogenic Si stocks?

(ii) To which extent are spatial patterns of *C. epigejos* and *P. australis* and corresponding Si stocks driven by initial soil properties?

(iii) What are the benefits and limitations of UAS-based remote sensing of phytogenic Si stocks?



## 2 Methods

### 2.1 Study Area

The artificial catchment "Chicken Creek" (6 ha in size, thereof 5.3 ha fenced) was constructed in an open-cast mining area of Lusatia, Germany (51.6049° N, 14.2667° E) in 2004–2005 (construction finished in September 2005). A 2-4 m thick, surficial layer of Quaternary, sandy sediments was dumped on a 1–2 m pan-shaped layer of Tertiary clays, which seals the whole catchment at its base. In the lower part of the catchment, additional clay dams were constructed on top of the clay layer (transverse to the slope). These dams act as a stabilization barrier preventing the sandy substrate from sliding downhill and

serve as a funnel to direct groundwater flow into the artificial pond downstream (Gerwin et al., 2010). Due to the artificial construction, the lower boundary conditions of the catchment site are clearly defined including knowledge about the 3D sediment structures (Gerke et al. 2013, Schneider et al., 2011). The construction work left a bare land surface on which natural vegetation could develop without disturbance (natural succession) but also created a zonal pattern of soil properties caused by the natural heterogeneity of the parent material taken from different areas in the fore-field according to the progression of the

mine (Gerwin et al., 2009) (Fig. 1).

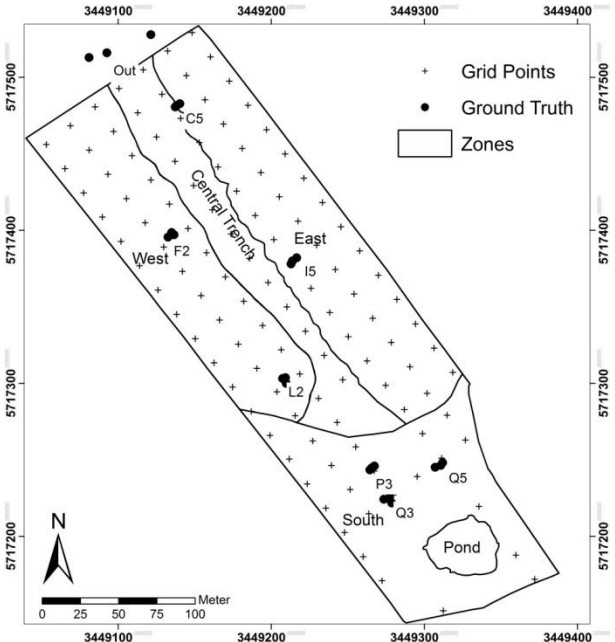

**Figure 1: Schematic map of the artificial catchment "Chicken Creek" showing grid points, ground truth sites and zones, delineating**
**areas with slightly different soil properties.**

*C. epigejos* has been present since the very beginning and belonged to the most dominating species since 2010, especially in the western part of the catchment. *P. australis* has also been present from the beginning of vegetation development in 2006,



but it was mainly restricted to the area around the pond in the southern part of the catchment (Elmer et al., 2011, Schaaf et al., 2010). The sub-continental climate is characterized by a mean annual precipitation of 563 mm and a mean annual temperature
of 8.9 °C.

## 2.2 Ground-based measurements

### 2.2.1 Aboveground biomass

Shoot biomass (include stems, leaves and inflorescence) with a dominance of *C. epigejos* was sampled at sites C5, F2, I5 and L2, which represents raster points equipped for an extensive soil moisture and temperature monitoring. Three additional sites
(CA1 to CA3) were sampled outside of the fenced area to include sites of high population densities (Fig. 1). At each site plants were cut within an area of 50 cm x 50 cm (0.25 m²) from three locations, which represent a (local) gradient of low, medium and high population densities. Analogous, 3 sites close to raster points P3, Q2 and Q5 with a dominant occurrence of *P. australis* were sampled. The dead, tufted biomass of *C. epigejos* and the brown shoots including the litter of *P. australis* where sampled separately within the same areas. This material of the preceding seasons will be referred to as litter hereafter. A subset
of the brown *P. australis* shoots have been retained for further analysis in order to find out whether the Si content is different from green shoots. All plant samples were oven dried over 48 h at 105 °C for further determination of Si content.

### 2.2.2. Si analysis of plant material

An aliquot of 5 g from the dried plant material was used to determine Si content. The collected plant material was carefully washed with distilled water to remove adhering soil particles and oven-dried at 45°C. Subsequently the samples were milled
using a knife mill (Grindomix GM 200, Retsch) in two steps: 4.000 rpm for 1 min. and then 10.000 rpm for 3 min. Sample aliquots of approximately 100 mg were digested under pressure in PFA digestion vessels using a mixture of 4 ml distilled water, 5 ml nitric acid (65%), and 1 ml hydrofluoric acid (40%) at 190 °C using a microwave digestion system (Mars 6, CEM). A second digestion step was used to neutralize the hydrofluoric acid with 10 ml of a 4%-boric acid solution at 150 °C. Silicon was measured by ICP-OES (ICP-iCAP 6300 Duo, Thermo Fisher Scientific GmbH) with an internal standard. To avoid
contamination, only plastic equipment was used during the complete procedure. Analyses were performed in three lab replicates.

### 2.2.3. Soil sampling and soil analysis

Soil sampling of the upper 30 cm was carried out subsequent to catchment completion in 2005 at 124 grid points in a 20 m x 20 m grid (see Fig. 1). Soil samples were analysed on various physicochemical soil properties (for details see Gerwin et al.,
2010). From these we used data of clay and nitrogen (N), which are known as important drivers for vegetation development at Chicken Creek (Elmer et al., 2013, Zaplata et al., 2011a, b). In addition, we analysed plant available potassium (K), phosphorus





(P), and Si content in retained samples from 2005 to analyse their effects on spatial patterns of *C. epigejos* and *P. australis* and corresponding aboveground phytogenic Si stocks.

Plant available K and P were determined with the double-lactate method is used in Germany for the determination of plant
available potassium and phosphorus. The extraction solution comprises a 0.04 m calcium lactate solution buffered with 0.02 m hydrochloric acid at pH of 3.6 using a soil-to-solution ratio of 1:50 (VDLUFA, 1991). Four grams of air dried soil (<2 mm) were weighed into polyethylene laboratory bottles, 200 mL of extraction solution were added and placed on a mechanical shaker for 90 min. After filtration the phosphorus concentration was determined by colorimetry (Gallery Plus, Microgenics) and the potassium concentration was measured using Flame-Atomic absorption spectroscopy (AAS-iCE 3300, Thermo
Fischer). The reported values are in mg K or P per 100 g dry soil.

Plant available Si (water-extractable Si, cf. Sauer et al., 2006) was determined as described in Puppe et al. (2017). In short, ten grams of dry soil (<2 mm) was weighed, put into 80 mL centrifuge tubes, and 50 mL distilled water was added with three drops of a 0.1% $NaN_3$ solution to prevent microbial activity. Total extraction time was 7 days. Twice a day tubes were gently shaken for 20 s by hand to prevent abrasion of mineral particles from colliding during constant (mechanical) shaking by using,
e.g., a roll mixer. After extraction solutions were centrifuged (4000 rpm, 20 min), filtrated (0.45 μm polyamide membrane filters), and Si was measured via ICP–OES (ICP-iCAP 6300 DUO, Thermo Fisher Scientific Inc). Only plastic equipment was used during the complete extraction procedure to exclude any Si contamination. Analyses of water-extractable Si were performed at a minimum of two lab replicates per sample.

The alkaline extractant Tiron ($C_6H_4Na_2O_8S_2 \cdot H_2O$) was used for the detection of potential differences in the amorphous silica
stocks (as a proxy of synthesized biogenic and pedogenic siliceous structures representing the main source for plant available Si), although a partial dissolution of primary minerals is well known (Sauer et al., 2006). However, due to the fact that the suitability of the so-called DeMaster technique (which represents the de facto standard method) for quantification of amorphous biogenic Si has been questioned recently (Meunier et al., 2014; Li et al., 2019), we performed no time-course extraction, but used a short time extraction (1 h) for all samples. Based on the short extraction time of only one hour we
excluded extensive extraction of mineral Si forms (cf. Kaczorek et al., 2019). The Tiron extractable Si ($Si_{Tiron}$) was determined by the method developed by Biermans and Baert (1977), modified by Kodama and Ross (1991). The extraction solution was produced by dilution of 31.42 g Tiron with 800 mL of distilled water, followed by addition of 100 mL sodium carbonate solution (5.3 g $Na_2CO_3$ + 100 mL distilled water) under constant stirring. The final pH of 10.5 was reached by adding small volumes of a 4M NaOH-solution. For the extraction 30 mg of dry soil were weighed into 80 mL centrifuge tubes and a 30 mL
aliquot of the Tiron solution was added. The tubes were then heated at 80°C in a water bath for 1h. The extracted solutions were centrifuged at 4000 rpm for 30 min, filtrated (0.45 μm polyamide membrane filters, Whatman NL 17) and Si concentrations measured by ICP-OES. To avoid contamination, only plastic equipment was used during the complete procedure. Analyses of Tiron extractable Si was performed in three lab replicates per sample.





### 2.3 UAS remote sensing

We used a fixed-wing UAS Carolo P360 (Fig. 2) with a wingspan of 3.6 m and a take-off weight of 22.5 kg. The UAS is equipped with a 12-band multi-camera array Mini-MCA 12 (MCA hereafter) (Tetracam Inc., CA, USA). The 12 narrow-band filters (between 10 nm and 40 nm bandwidth) cover the spectral range from visible to near infrared wavelengths with focus on the characteristic reflectance features of healthy vegetation (chlorophyll absorption band around 660 nm, the red-edge region between 680 nm and 750 nm) and one of the water absorption bands around 950 nm. Hereafter, bands will be denoted according

their respective center wavelength in nm (e.g. $b_{713}$ for the red edge band).

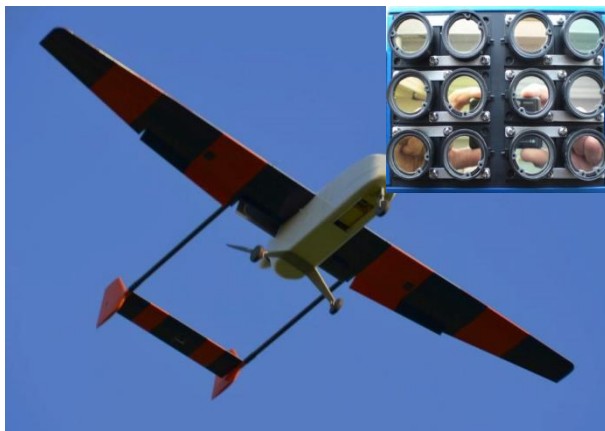

**Figure 2: Unmanned Aerial System (UAS) Carolo P360 with multi-camera array Mini-MCA 12 (visible lenses at the underside of the camera).**

The UAS mission was conducted during the flowering period of *C. epigejos* on 21 August 2014 under acceptable flight

conditions (moderate wind speed and little cloud shadow). A small negligible area was covered in the south-east part with no or little occurrence of *C. epigejos* and *P. australis* respectively. At this time, the foliage is medium green and the large inflorescences are clearly visible. This facilitates the (i) spatial delineation and (ii) spectral distinction between *C. epigejos* populations and dozens of other existing grass-like species. Due to camera specifications a unique flight altitude of 163 m above ground is required to achieve the desired ultrahigh ground sampling distance (GSD) of ~ 0.1 m. Details about UAS,

camera specifications and mission settings are presented in appendix A1. The post-processing chain including radiometric corrections, mosaicking and geo-referencing is described in more detail in appendix A2.

### 2.4 Image classification

A supervised ML classification was applied to identify individual dominant plant species or communities. The ML classifier requires a proper selected set of training areas for all objects visible in the image.. For this purpose we used field inspections

and available botanical mappings at grid points (data provided by M. Zaplata). Details regarding conditions and constraints of the ML classifier are given in appendix B1..



Finally 16 relevant land cover classes (LCC) were defined including three classes of non-vegetation (shadow, bare soil, open water) and four classes of legume and non-legume woods (Table 1). The statistical separability of the LCC signatures was computed before each classification run. Divergence, an often used separability measure in remote sensing, is computed using

the mean and variance-covariance matrices of the pixel values representing the training area. The Jeffries-Matusita (J-M) distance (Kavzoglu and Mather, 2000), the divergence measure used in this study, was computed for all possible LCC pairs. A computed value of zero indicates that classes are inseparable and a value of 1414 means total separability (Swain, 1978).

**Table 1: List of the 16 relevant land cover classes (LCC) as predefined for classification on basis of field survey and botanical mappings.**

| Group | LCC No | | Description |
|---|---|---|---|
| Grass-like communities | 01 | *C. epigejos* - d | Dense population |
| | 02 | *C. epigejos* - dt | Dense population close to trees (sunlit side of *Robinia pseudoacacia*) |
| | 03 | *C. epigejos* - m | Population with a minor fraction of visible Herbs, Mosses and Lichens |
| | 04 | *C. epigejos* - HML | Sparse population with a large fraction of visible Herbs, Mosses and lichens |
| | 05 | *P. australis* - d | Dense population with a particular fraction of shadow |
| | 06 | *P. australis* - m | Population with a minor fraction of other visible grass-like species and shadow |
| | 07 | *P. australis* – HML | Sparse population with a large fraction of visible Herbs, Mosses and lichens |
| | 08 | *F. rubra* | Population of *Festuca rubra* agg. |
| Legume-, non-legume Herbs; Mosses; Lichens (HML) communities | 09 | HML-0 | Herbs, Mosses and lichens populations without any other visible grass-like species |
| | 10 | HML-1 | Herbs, Mosses and lichens populations with a minor fraction of grass-like species (unidentified) |
| Legume-, non-legume Woods individuals | 11 | *R. pseudoacacia/S. caprea* | Large individuals of *Robinia pseudoacacia* and *Salix caprea* |
| | 12 | *H. rhamnoides* | Large individuals of *Hippophae rhamnoides* |
| | 13 | *P. sylvestris* | Small individuals of *Pinus sylvestris* |
| Non - vegetation | 14 | Shadow | Shadow of trees and bushes |
| | 15 | Bare soil | Predominately sandy substrate |
| | 16 | Open water | |


## 2.5 Calculation of the Normalized Difference Vegetation Index (NDVI)

The NDVI is an intrinsic vegetation index that simply accounts for the chlorophyll absorption feature in the red (R) and the structural information inherent in high NIR reflectance of a green vegetation canopy. It does not involve any external factor other than the measured spectral information and is calculated by:

$$NDVI = \frac{(NIR - R)}{(NIR + R)} \tag{1}$$

where NIR and R are the reflectance in the near-infrared and red band. Since radiometric calibration was renounced in this study, NIR and R refer to DNs instead of at-surface reflectance. In order to investigate the potential of the available bands in the red edge (RE, the steep incline between VIS and NIR reflectance typically for green vegetation) and NIR domain, five variations of the NDVI were calculated. The four NIR bands correspond to $b_{831}$, $b_{861}$, $b_{899}$ and $b_{953}$. In case of the RE band $b_{713}$,

NIR have to be substituted by RE in Eq. 1.





Coefficients of determination (R²s) were calculated for the relationships between the examined VIs and the ground measured fresh shoot biomass of *C. epigejos* and *P. australis* as individual species and for the pooled dataset consisting of all measures. Since the fresh litter accumulated during the preceding seasons is covered by the tall growing *C. epigejos* and *P. australis* and therefore cannot be estimated directly, R²s were calculated analogous for the sum of the fresh shoot biomass and litter. The

latter was done in order to find out whether correlations between calculated VIs and fresh shoot biomass at the one side, and - on the other hand-, between VIs and fresh shoot biomass including the litter can be used for an indirect estimation of fresh litter.

**2.6 Statistical analysis**

Correlations were analysed using Spearman's rank correlation ($r_s$). Significances between two independent samples were

verified with the Mann–Whitney U test. Significances between more than two independent samples were tested with the Kruskal-Wallis analysis of variance (ANOVA) followed by pairwise multiple comparisons (Dunn's post hoc test). Statistical analyses were performed using software package SPSS Statistics (version 22.0.0.0, IBM Corp.).

**3 Results**

**3.1 Image classification**

The computed J-M distance for the 120 LCC pairs shows an average of 1386 and a minimum of 1142. While the signatures of 79 LCCs are almost totally separable (J-M distance ≥1 410), a fairly separability exists for 32 LCC pairs (J-M distance ≥ 1248). Values below 1249 were computed for 9 LCC pairs indicating a poor separability.

Figure 3 depicts the mean signatures of the 16 LCCs. In Fig. 3a all LCCs others than those classified as *C. epigejos* or *P. australis* (8 to 16) reveal large differences and can thus be clearly distinguished. While open water and shadow shows the

lowest DNs over all spectral bands, the highest DNs in the visible range are characteristic for sandy soil (Stoner and Baumgarder, 1982). Regardless the use of DNs instead of reflectance the signatures of trees and bushes show the characteristic features of green vegetation. With decreasing chlorophyll content and population densities of photosynthetically active species (here *F. rubra*) and a simultaneously increasing fraction of shadow, cryptogam species and dead plant material (LCCs 9 and 10), DNs show a more or less monotonous increase of DNs from the VIS to the NIR wavelength domain. The signatures of

LCCs 1 to 7 are depicted in Fig. 3b.The signature of the dense *P. australis* population (LCC 5) exhibits a shape similar to the signatures of trees and bushes but DNs are lower in the NIR domain caused by a higher fraction of shadow visible to the sensor. All other signatures appear similar regarding the general shape (monotonous increase) but the separability is fairly high for most of them caused by the lower overall reflectance and the flatter slope of the signature with an increasing fraction of visible non-photosynthetic material and shadow. Amongst the 10 pairs showing poor separability, 4 pairs comprise LCCs describing

transitional states within the same species (e.g. *P. australis*; LCC 6 and LCC 7) or cannot clearly distinguish between *C. epigejos* and *P. australis*. The first case is uncritical since the respective LCCs will be treated as one LCC in the further





analysis. The second case cannot be avoided due to co-occurrence within the same area of interest. However resulting misclassifications are negligible with respect to the small differences of Si content observed for both species.

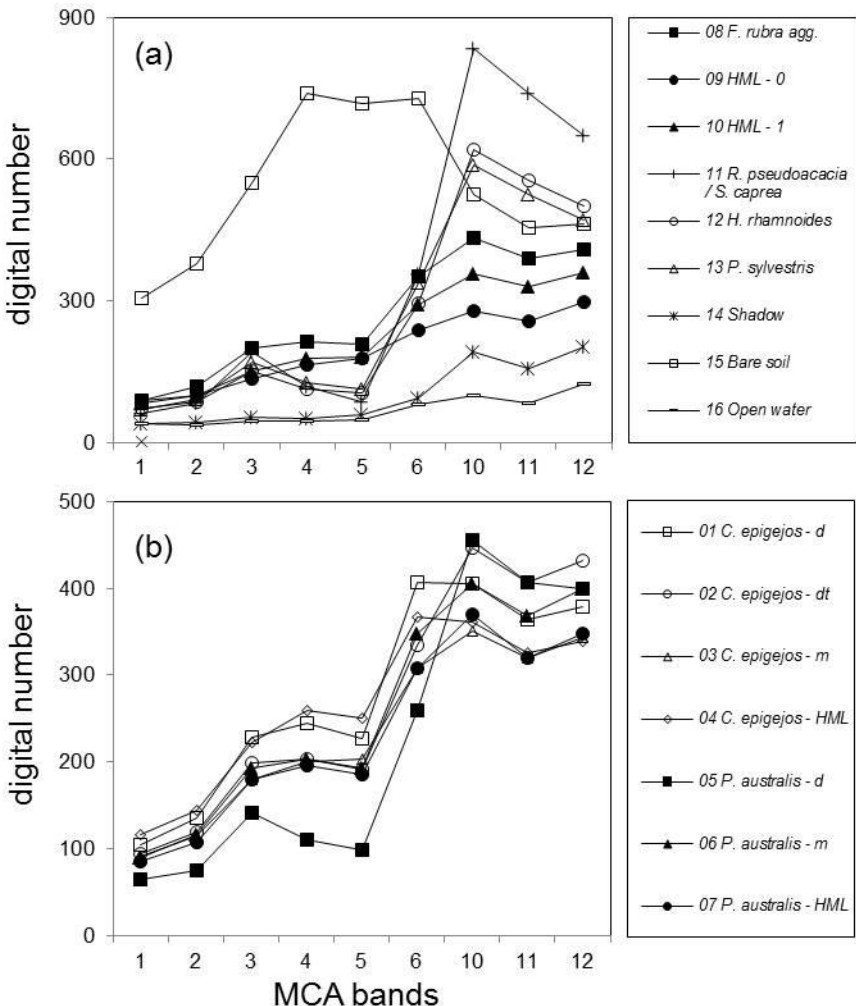


**Figure 3: Mean signatures of classified LCCs others than C. epigejos and P. australis (a) and respective signatures of C. epigejos and P. australis classes (b). Note the different scaling of the y-axes.**

Constraints have to be accepted regarding the poor separability between *F. rubra* (LCC 8), *C. epigejos* (LCC 2) and *P. australis*

(LCC 6 and 7) respectively. Although mean signatures indicate fairly separabilty at least in the NIR domain, class variances are large and diminish the separability.



## 3.2 Spatial coverage and zonal distribution of land cover classes

*C. epigejos* (40 %) and *P. australis* (22 %) cover most of the area followed by trees, bushes and the respective shadow with a spatial coverage of 19 %. Legume and non-legume herbs, mosses and lichens without or with a minor fraction of grass-like
species cover 4.5 % and 4.8 % respectively. The remaining area was classified as F. rubra (LCC 8; 3.5 %). open water (3.1 %) and bare soil (2.6 %). The spatial coverage of each of the 16 LCCs is depicted in Fig. 4.

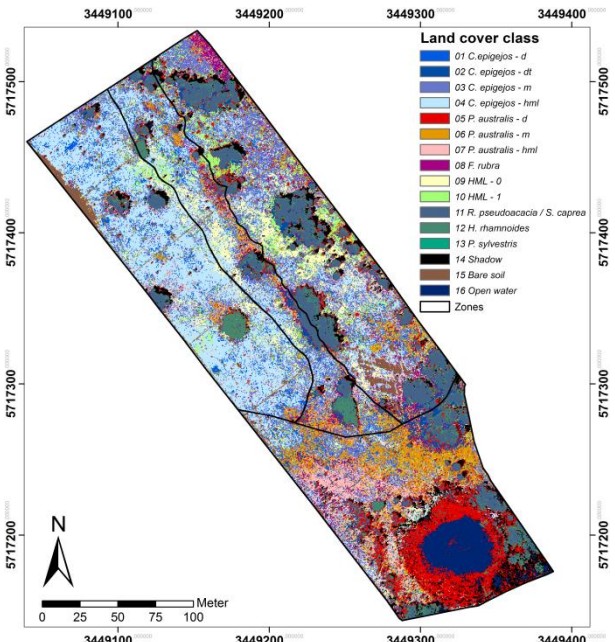

**Figure 4: Spatial distribution of classified land cover classes (LCC).**


The Si accumulators *C. epigejos* and *P. australis* show a clear zonal distribution. *C. epigejos* is widespread in the northern zones of the catchment. Populations with high density (LCC 1 and 2) occur regularly nearby the sunlit side of R. pseudoacacia (LCC 2). In reality the populations are distributed more circular around the trees but the shaded side during image acquisition prevented from classification. Other high density populations (LCC 1) are predominately spread as smaller patches in the
western zone. A clear zonal distinction can be observed for the two *C. epigejos* classes dominating the northern zones. The sparse populations (LCC 4) with a spatial coverage of 19 % occur in the western zone and the central trench. LCC 3 as the second largest class (15 %) is widespread in the eastern zone and the central trench, but spatially separated by trees, bushes and larger patches of other LCCs (HML 0 and 1, bare soil). *P. australis* mainly occurs in the southern zone of the catchment with a dense population concentrated around the artificial pond. A narrow band runs from the north-west to the south-east
along the central trench. The less dense and the sparse populations (LCCs 6 and 7) occur in a band-like pattern running in west-east direction. The zonal pattern reflects mainly wetter areas, i.e. the pond's fringe or sites where lateral groundwater





flow approaches the surface (return flow). The narrow NW-SE band marks the edge of an erosion gully along the central trench and the W-E band matches the belowground clay dams, where groundwater flow is forced towards the land surface.

### 3.3 Ground-based measurements of above ground biomass of *C. epigejos* and *P. australis*

With the exception of the separately sampled litter the variation of fresh shoot biomass is larger than the respective variation at *P. australis* sites (Table 2). The lowest amount was sampled at plot I5-2 with 126 g m$^{-2}$. In contrast, amounts at plot CA-1 are eight times higher (1018 g m$^{-2}$). Since the sampled litter is predominately dead material, differences between fresh and dry samples are naturally small. However, the overall variance is similar to that of fresh and dry shoot biomass but it should be noted that no correlation exists between the two quantities. High amounts of litter were sampled at plots with low and high

shoot biomass (e.g. I5-2 and CA-2 respectively). The variation between the three plots at one site reflects the small-scale variability of the island-like populations of *C. epigejos* with decreasing shoot biomass from the center to the margins. While amounts at L2-3 (center) are 1.5 times higher than at L2-2 (margin) the factor increases to almost 3 between plots I5-3 and I5-2. The separately sampled litter varies within a range between 14 g m$^{-2}$ (L2-1) and 126 g m$^{-2}$ (I5-3).

**Table 2: Mean, standard deviation (SD), min, max and Coefficient of Variation (CV) calculated for fresh and dry biomass samples**

**of *C. epigejos* and *P. australis*.**

| | *C. epigejos* (n = 15) | | | | *P. australis* (n = 9) | | | |
|---|---|---|---|---|---|---|---|---|
| | Fresh | | Dry | | Fresh | | Dry | |
| | green shoot | litter | green shoot | litter | green shoot | litter (incl. brown shoot) | green shoot | litter (incl. brown shoot) |
| Mean | 387 | 84 | 214 | 78 | 456 | 634 | 204 | 482 |
| Sd | 243 | 39 | 131 | 36 | 262 | 332 | 120 | 310 |
| Min | 126 | 14 | 57 | 13 | 133 | 251 | 50 | 178 |
| Max | 1018 | 150 | 465 | 139 | 830 | 1337 | 384 | 1106 |
| CV [%] | 63 | 47 | 61 | 47 | 58 | 52 | 59 | 64 |

Since there was no access to the dense *P. australis* population in the surrounding of the pond, highest and lowest amounts of fresh and dry shoot biomass were collected at site Q5. The small-scale variability in fresh green shoot biomass (factor 4.9) is even higher compared with *C. epigejos*. The fresh and dry samples of litter show a high variance indicated by a CV of 74 %

between plots. Again lowest and highest amounts were sampled at site Q5 with fresh weights of 57 g m$^{-2}$ at Q5-2 and 646 g m$^{-2}$ at Q5-1.

### 3.4 VI performance

All relationships calculated for the combinations between NDVIs and ground measured biomass show a positive linear trend. The low NDVI values, ranging between 0.08 (*C. epigejos*; L2-2; NDVI$_{b891}$) and 0.41 (*P. australis*; P3-2; NDVI$_{b856}$) indicate





no saturation effects. The respective R²s, Root Mean Square Errors (RMSE) [g m$^{-2}$] and Mean Relative Errors (MRE) [%] for the examined relationships are summarized in Table 3.

Regarding the band combinations the NDVI using the RE b$_{713}$ (Fig. 5) was found the best predictor for both species as well as of fresh green shoot biomass (Fig. 6a) and the sum of fresh green shoot biomass and fresh litter (Fig. 6b). In both cases high R²s can be observed for *C. epigejos* (R² = 0.87) and for *P. australis* (R² = 0.74 and R² = 0.78 respectively). While the other

band combinations perform well for *C. epigejos*, the predictive power decreases significantly for *P. australis* (R²s range between 0.40 and 0.11). VIs of *P. australis* show higher values at same fresh green biomass amounts and generally a larger scatter of values than of *C. epigejos*. This effect is related to the different plant architecture of the two species. The combination of broader green leaves and larger but less individual plants of *P. australis* leads to higher VIs representing the same biomass on the one hand. On the other hand the number of pixel affected by shadow with decreasing population densities increases the

scatter of VI values.

**Table 3: R², RMSE and MRE for relationships between examined NDVIs and aboveground biomass of *C. epigejos* and *P. australis*.**

| | fresh green shoot biomass | | | | | | fresh green shoot biomass plus litter | | | | | |
| | *C. epigejos* | | | *P. australis* | | | *C. epigejos* | | | *P. australis* (litter incl. brown shoot) | | |
| Band | R² | RMSE | MRE | R² | RMSE | MRE | R² | RMSE | MRE | R² | RMSE | MRE |
| | | [g m-2] | [%] | | [g m-2] | [%] | | [g m$^{-2}$] | [%] | | [g m-2] | [%] |
|---|---|---|---|---|---|---|---|---|---|---|---|---|
| b$_{713}$ | 0.87 | 84 | 27.3 | 0.74 | 126 | 38.5 | 0.87 | 89 | 19.5 | 0.79 | 353 | 31.3 |
| b$_{861}$ | 0.82 | 100 | 29.2 | 0.37 | 197 | 55.3 | 0.81 | 106 | 20.9 | 0.36 | 389 | 30.0 |
| b$_{899}$ | 0.82 | 99 | 31.6 | 0.33 | 202 | 60.6 | 0.80 | 108 | 23.1 | 0.40 | 376 | 29.1 |
| b$_{953}$ | 0.73 | 123 | 36.6 | 0.11 | 234 | 76.6 | 0.70 | 132 | 27.1 | 0.21 | 434 | 39.7 |



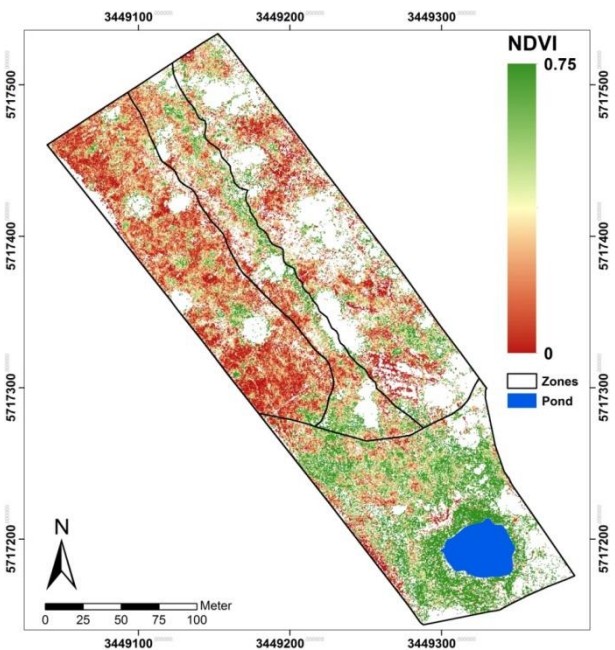

**Figure 5: NDVI$_{b713}$ calculated from Eq. 1 using the red edge band. The area corresponds with the classified spread of C. epigejos (LCCs 1 – 4) and P. australis (LCCs 5 – 7).**

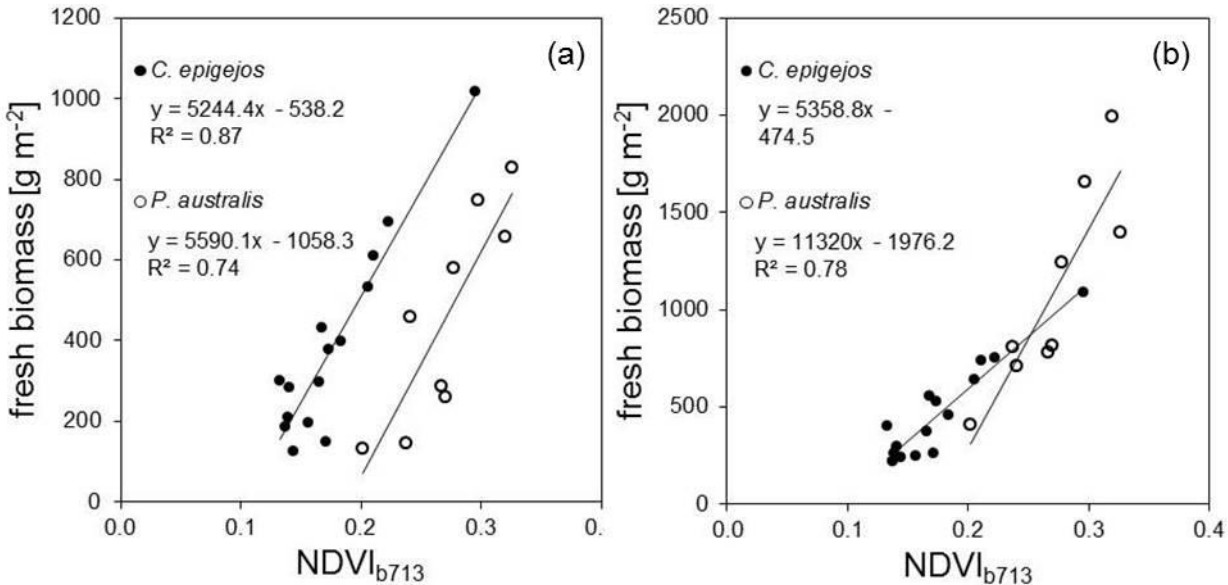

**Figure 6: Respond of NDVIb713 to fresh green shoot biomass (a) and the sum of fresh green shoot biomass and litter (b) of C. epigejos and P. australis.**





### 3.5 Estimation of dry biomass fractions

Taking into account the results of section 3.3, dry green shoots and the sum of dry green shoots and litter were calculated separately for each species. The quantities are highly correlated with R²s of 0.88 and 0.86 for *C. epigejos* and 0.99 and 0.97 for *P. australis*. The respective linear regressions (Fig. 7a and 7b) were used to model their spatial distribution in a first step.

In a second step, the modelled amounts of dry green shoots were subtracted from the modelled sum of dry green shoots and litter to yield the spatial distribution of dry litter.

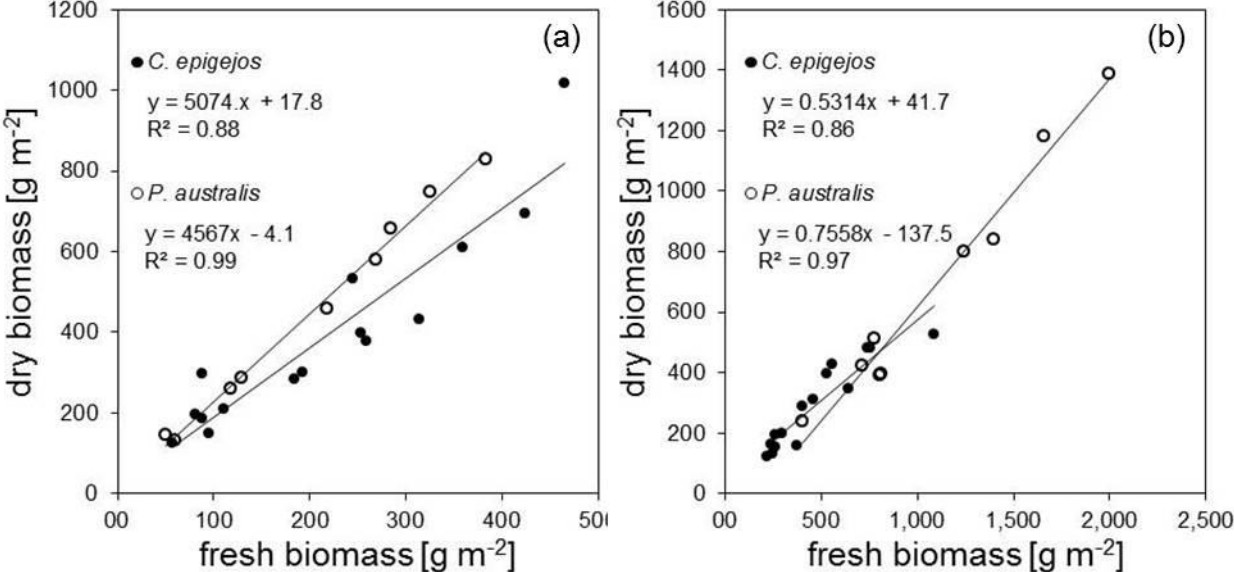

**Figure 7: Correlation between fresh and dry green shoot biomass (a) and the respective biomass including litter (b) for C. epigejos**
**and P. australis.**

### 3.6 Silicon content and stocks of *C. epigejos* and *P. australis*

The mean Si content within dry litter of *C. epigejos* (3.7 %) is 1.8 times higher than the Si content in the dry shoot biomass of the current year (2.1 %). The effect is less pronounced for the three analyzed fractions of *P. australis* (Table 4). The Si content range between 3.0 % (dry litter without dry brown shoot biomass), 2.5 % (dry brown shoot biomass) and 2.3 % (dry green

shoot biomass). As we regard the sum of the first two fractions as litter, the mean of both contents (2.7 %) was used for further calculations. The mean fairly represents real conditions since both fractions, on average, contribute to the sum in equal amounts. Finally, the respective Si content was used to calculate the Si stocks in both fractions of *C. epigejos* and *P. australis*. The areas given in Table 4 equal the summed areal coverage of the classified LCCs 1–4 (*C. epigejos*) and 5–7 (*P. australis*). The total Si stock (sum of green shoot biomass and litter) accumulated in *P. australis* contributes to 64 % (275 kg) to the total

Si stock calculated for the whole catchment (429 kg) despite the fact that the areal coverage is almost half that of *C. epigejos*.





**Table 4: Mean Si content of biomass fractions of *C. epigejos* and *P. australis* and respective Si stocks calculated from the areal coverage derived from ML classification.**

| | | Si content [%] | | Si stocks | | |
| --- | --- | --- | --- | --- | --- | --- |
| | Fractions | Mean | CV [%] | [g m$^{-2}$] | Area [m²] | Si [kg] |
| *C. epigejos* | shoot biomass | 2.1 (0.7) | 33 | 6.0 | 20755 | 125 |
| n = 15 | litter | 3.7 (1.0) | 27 | 1.4 | | 30 |
| | total | | | 7.4 | | 155 |
| *P. australis* | shoot biomass | 2.3 (0.4) | 17 | 7.5 | 10063 | 75 |
| n = 9 | litter | 2.7 (0.4) | 13 | 19.8 | | 199 |
| | total | | | 27.3 | | 274 |
| Catchment | | | | 13.9 | 30818 | 429 |

This is simply caused by the fact that *P. australis* forms more dry biomass per unit ground area than *C. epigejos* (factor 2.3 on average). The spatial distribution of total Si stocks calculated on the basis of the real areal coverage of both species is depicted in Fig. 8a and Fig. 8c. According the higher biomass production, Si stocks of *P. australis* reach a maximum (98 g m$^{-2}$) in the fringe around the pond. The majority between 3 g Si m$^{-2}$ and 60 g Si m$^{-2}$ occurs in the southern zone and in a narrow band along erosion gullies in the central trench.





**Figure 8: Spatial distribution of Si stocks of C. epigejos (a) and P. australis (c) and total amounts of Si accumulated in the four zones by C. epigejos (b) and P. australis (d). Note the different legend scales in (a) and (c).**

With the exception of few patches in the southern zone the vast majority of values do no not exceed 17 g m$^{-2}$ in the case of *C. epigejos*. It is noticeable that the southern zone shows a clear two-parted internal zonation. The northern part is a type of a transition zone showing a co-occurrence of both species with high Si accumulation in *C. epigejos*, whereas the southern part is dominated by *P. australis* in the surrounding of the pond. This indicates that the spatial pattern is caused by soil moisture conditions rather than the initial spatial differences in soil properties induced by construction work and explains the relative low zonal contribution of 17% (26 kg) to the total Si stock of 154 kg (Fig. 8b).

Regardless the accumulated Si in the southern zone and the central trench, the clear distinction between the eastern and western zone is striking. While the western zone contributes to 40% (62 kg), only 23% (35 kg) were accumulated in the eastern zone. Values correspond to mean Si stocks of 4.0 g m$^{-2}$ and 2.3 g m$^{-2}$ respectively.





Figure 8d shows the Si stocks accumulated in *P. australis* calculated for each of the four zones. While the southern zone contributes to 66 % (180 kg Si) to the total amount, Si stocks in northern zones play a minor role. Mean Si stocks reach 12.2 g Si m$^{-2}$ in the southern zone but significantly lower values in the eastern (3.1 g Si m$^{-2}$) and the western zone (1.0 g Si m$^{-2}$). Although mean Si stocks in the central trench are similar to those in the eastern zone (3.6 g Si m$^{-2}$) the occurrence of *P.*
*australis* tends to be more linear along the main gully. Regarding habitat requirements of *P. australis*, the occurrence is a result of higher soil moisture conditions in this area and the partially wet and flooded sites around the pond.

**3.7 Relationship between Si stocks of *C. epigejos*, *P. australis* and site properties**

From a total of 124 grid points we excluded all locations in the central trench due to disturbances in soil properties and related nutrient availability caused by water erosion compared to the initial conditions in 2005. Moreover, we excluded several grid
points within the remaining zones located below trees and bushes without any classified occurrence of *C. epigejos* and *P. australis*. Finally, Si stocks and site properties of 35 grid points located in the western, 32 in the eastern and 18 in the southern zone were used for statistical analysis. Mean Si stocks were extracted from a squared area of 25 m² around each grid point.

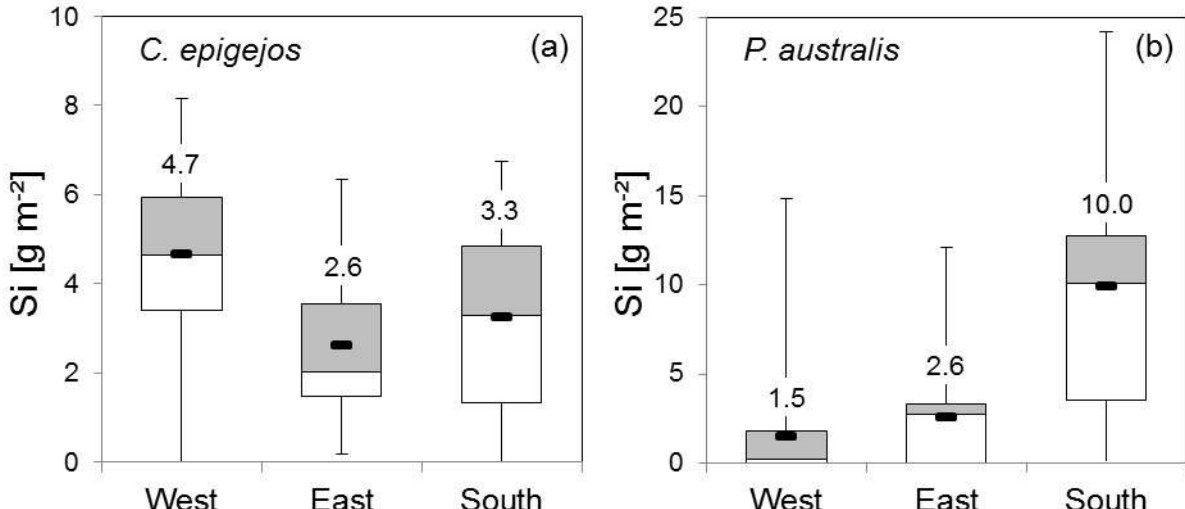

**Figure 9: Distribution of Si stocks of C. epigejos (a) and P. australis (b) at grid points in the western, eastern and southern zone.**
**Note the different scaling of the y-axes.**

Compared to the zonal means of 4.0 g m$^{-2}$, 2.3 g m$^{-2}$ and 1.8 g m$^{-2}$ for *C. epigejos*, the Si stocks extracted at grid points increased to 4.7 g m$^{-2}$ in the western, 2.6 g m$^{-2}$ in the eastern and 3.3 g m$^{-2}$ in the southern zone (Fig. 9a). However the ratios of 0.9 for the western and eastern zone indicate an adequate representation of the grid points for both zones. The large difference in the southern zone is mainly caused by the two-parted zonation with almost no occurrence of *C. epigejos* in more
than half of the southern zone area. Zonal Si stocks accumulated in *P. australis* in the western, eastern and southern zone (1.0 g m$^{-2}$, 3.1 g m$^{-2}$ and 12.2 g m$^{-2}$) are sufficiently represented by grid points in all zones (1.5 g m$^{-2}$, 2.6 g m$^{-2}$ and 10.0 g m$^{-2}$) (Fig. 9b).



Among the examined soil properties, means of clay content (Fig. 10a) show a corresponding trend with respect to Si accumulation in dry biomass of *C. epigejos* for all three zones.

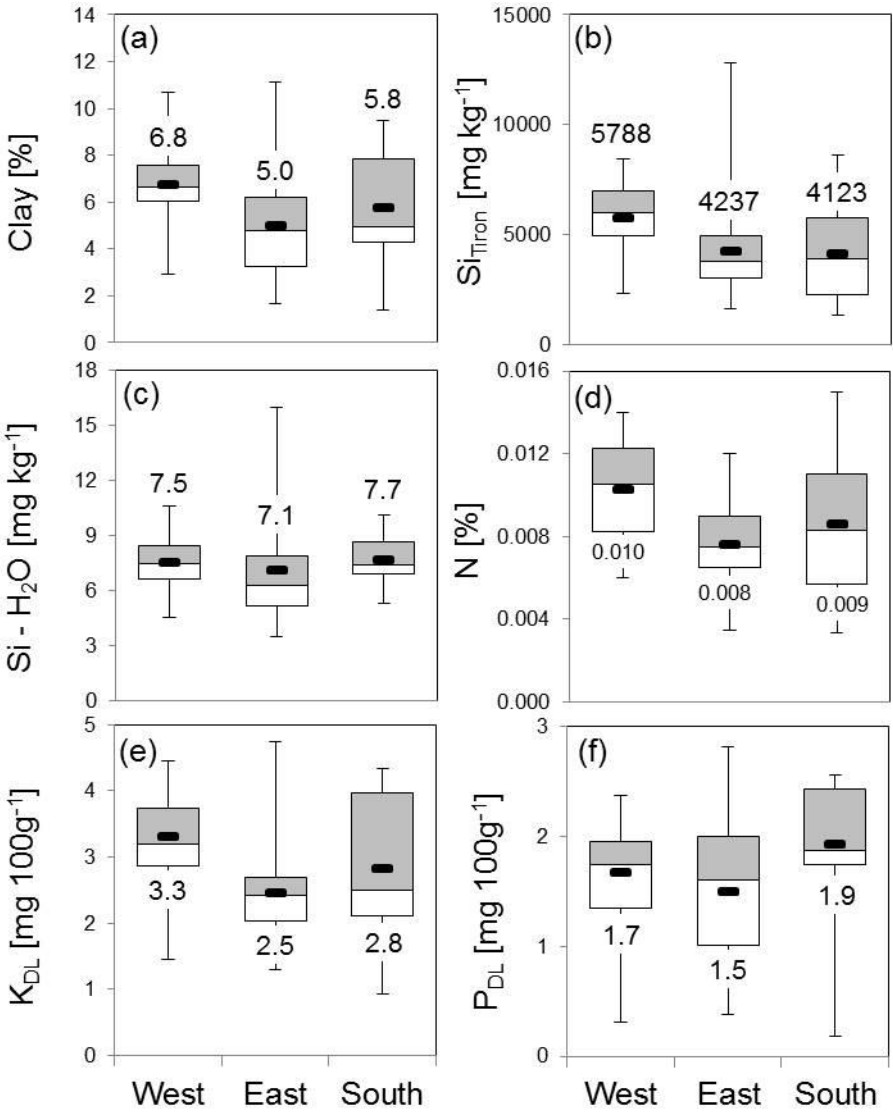


**Figure 10: Distribution of soil properties and nutrients at grid points in the western, eastern and southern zone. Figures depict the data distribution of clay content (a), Tiron extractable amorphous Si (SiTiron) (b), water soluble Si (Si – H₂O) (c), nitrogen (d), potassium (e), and phosphorus (f).**

Significant differences were found for $Si_{Tiron}$ (Fig. 10b) between the western and the eastern zone, whereas values in the

southern zone do not differ very much from those in the eastern zone. The water soluble Si (Fig. 10c) can be ruled out as a driving factor since there is no differentiation between means over all zones. The zonal distribution of the most important nutrients draws a similar picture. Highly significant differences of means between the western and eastern zone were found for nitrogen (Fig. 10d) and potassium (Fig. 10e). For both nutrients, slightly higher values compared to the eastern zone can





be observed for the southern zone which is in accordance with higher Si stocks. It can be considered that $P_{DL}$ is not a driving

factor for the spatial spread of *C. epigejos* populations, since availability is more or less equal in all three zones (Fig. 10f). As stated before, the occurrence of *P. australis* is governed by soil moisture conditions. Therefore neither examined soil properties nor nutrient availability show a recognizable impact on the spatial spread along zones. However, we cannot exclude the possibility of relationships within zones on the basis of grid points especially in the two-parted southern zone.

## 4 Discussion

### 4.1. Aboveground phytogenic Si stocks at Chicken Creek

Mean aboveground Si stocks of *P. australis* and *C. epigejos* are surprisingly high and are comparable to or markedly exceed reported values for the Si storage in aboveground vegetation, e.g., in the Great Plains (short grass steppe and tall grass prairie, 2.2 to 6.7 g Si m$^{-2}$, Blecker et al., 2006), the tropical humid grass savanna (tall grass *Loudetia simplex*, 3.3 g Si m$^{-2}$, Alexandre et al., 2011) or forested biogeosystems (beech forest: 8.3 g Si m$^{-2}$, Sommer et al., 2013; Beech-fir forest: 18 g Si m$^{-2}$, pine

forest: 9 g Si m$^{-2}$, Bartoli, 1983). Maxima of Si stocks at Chicken Creek reach values (up to 98 g Si m$^{-2}$) that are comparable to the Si storage in wetlands (50 to100 g Si m$^{-2}$, Struyf and Conley, 2009). Due to the fact that the Si content of *C. epigejos* and *P. australis* are in line with published values for grasses in general (Hodson et al., 2005), we conclude the observed Si stocks to be predominantly driven by the (high) biomasses of both plants. If we assume a more or less steady annual Si accumulation in *C. epigejos* for 5 years (*C. epigejos* became one of the most dominating plant species since 2010, Zaplata et

al., 2011b), the mean Si accumulation in the aboveground biomass of *C. epigejos* amounted to about 6.0 g Si m$^{-2}$ per year. If we further assume a similar time span for the Si accumulation in *P. australis*, the mean Si accumulation in the aboveground biomass of *P. australis* amounted to about 7.5 g Si m$^{-2}$ per year. Thus, annual Si fixation in *C. epigejos* and *P. australis* at Chicken Creek exceeds published data on annual biosilicification rates of temperate forest biogeosystems (beech forest: 3.5 g Si m$^{-2}$ yr$^{-1}$, Sommer et al., 2013; Beech-fir forest: 2.6 g Si m$^{-2}$ yr$^{-1}$, pine forest: 0.8 g Si m$^{-2}$ yr$^{-1}$, Bartoli, 1983; Douglas fir

forest: 3.1 g Si m$^{-2}$ yr$^{-1}$, Norway spruce forest: 4.4 g Si m$^{-2}$ yr$^{-1}$, black pine forest: 0.2 g Si m$^{-2}$ yr$^{-1}$, European beech forest: 2.3 g Si m$^{-2}$ yr$^{-1}$, oak forest: 1.9 g Si m$^{-2}$ yr$^{-1}$, Cornelis et al., 2010).

In the light of potential aboveground biomasses of, e.g., *C. epigejos* (up to about 700 g m$^{-2}$, Rebele and Lehmann, 2001), our results emphasize the significance of grasses for Si cycling in general. In this context, eutrophication is one of the most important drivers of the increased abundance of *C. epigejos* in many regions of Central Europe, especially East Germany,

Poland, and Czech Republic (Rebele and Lehmann, 2001), while on the other hand eutrophication might also be one of the drivers of the decline of *P. australis* in numerous European wetlands since the 1950s (Van der Putten, 1997). Considering the net primary production of the worldwide major biome types and the average amounts of Si fixed in the corresponding vegetation, the significance of grasses for Si cycling becomes much clearer: tropical woodland and savanna, temperate steppe, tundra, wetland, and cultivated land belong to the biome types where Si is actively accumulated and vegetation is widely

dominated by grasses (Carey and Fulweiler, 2012). Humans directly affect the distribution and size of these biomes and thus





influence corresponding Si cycling through intensified land use (forestry, agriculture) (Struyf et al., 2010, Vandevenne et al., 2015a, b). Si exports through harvested crops generally lead to a Si loss in agricultural used soils (= anthropogenic desilicification) (Desplanques et al., 2006, Guntzer et al., 2012, Keller et al., 2012, Meunier et al., 2008, Vandevenne et al., 2012). On a global scale, about 35 % of Si accumulated in vegetation is synthesized by field crops and this proportion is going
to increase with increased agricultural production within the next decades (Carey and Fulweiler 2016). In this context, targeted manipulation of Si cycling might be a promising strategy to enhance carbon sequestration in agricultural biogeosystems to mitigate climate change (Song et al., 2014).

### 4.2. Initial soil properties as drivers of spatial patterns of *C. epigejos* and *P. australis* and corresponding Si stocks

In general, plant biomass and its distribution is mainly controlled by climatic, edaphic (e.g., soil moisture/texture, pH, and
nutrients) and geographic-historic factors as well as by species interactions (e.g., consumption by herbivores) and (anthropogenic) perturbations (e.g., Polis 1999). At Chicken Creek consumption of plants by herbivores can be generally excluded, because the total study area is fenced. The composition and structure of plant communities thus is mainly governed by climatic and edaphic factors at Chicken Creek. Studies of Zaplata et al. (2011a, 2013) indicated that differences in vegetation dynamics at Chicken Creek can be directly derived from slight differences in edaphic conditions resulting from construction
work with large machines (Gerwin et al., 2010).

In this context, especially differences in soil pH, carbon content, calcium carbonate and conductivity between the sandier eastern and the loamier western part were identified to influence plant species distribution in general (Zaplata et al., 2013). Our results on hand generally corroborate this differentiation between zones of Chicken Creek with clay, N, $K_{DL}$, and Tiron extractable Si content as important drivers of the small-scale distribution of *C. epigejos*. Süß et al. (2004) analysed plant
successional trajectories and corresponding drivers in calcareous sand ecosystems in the northern upper Rhine valley in Germany. They found the successional trajectories of *C. epigejos* to be correlated to total N, extractable P and K as well as soil moisture. In contrast, the most important nutrients for aboveground biomass production of *C. epigejos* seem to be N and calcium, while P and K seem to have no significant effect on biomass production (Rebele and Lehmann, 2001). This also is in line with our observation that $P_{DL}$ seems to be no driver of the small-scale distribution of *C. epigejos* at Chicken Creek. Plant
available Si concentrations seem to be no driver for the distribution of *C. epigejos* and *P. australis* as well. This might be a hint that Si accumulation in plants is probably more influenced by the phylogenetic position of a plant than by environmental factors like temperature or Si availability (cf. Prychid et al., 2004, Hodson et al., 2005, Cooke and Leishman 2012).

### 4.3. Benefits and limitations of UAS-based remote sensing of phytogenic Si stocks

Natural ecosystems are characterized by an abundant flora, arranged in a complex spatial pattern. Thus, a sufficient
classification of all relevant species in an ecosystem is challenging or even impossible and has been addressed in previous studies (e. g. Dunford et al., 2009; Laliberte et al., 2011; Husson et al., 2016).





While larger individuals like trees and bushes are easy to identify, the size of the majority of species at Chicken Creek is far below the 10 cm spatial resolution of the multispectral imagery used in our study and tall growing plants or broad leaved species prevent the sensor of seeing low growing species below. As a consequence, and in contrast to the classification of

monocultures, the produced LCC map of the Chicken Creek catchment merely represents the spatial distribution of species or species compositions visible to the sensor. For this reason, we could hardly use the botanical mapping at grid points provided by M. Zaplata to validate our classification but we could use these data as a rough quality check. Hence training areas for the classification have been defined giving priority to the two main Si accumulators under study. However, both limitations were of minor importance in the case of *P. australis* and *C. epigejos*. Even smaller patches of both species were large enough to be

identified unless the spatial resolution. With the exception of the transition zone northeast of the pond, where *C. epigejos* co-exist below *P. australis*, populations are spatially separated and represent the uppermost layer of the canopy within the respective plant community. This may lead to slight underestimations of the spread of *C. epigejos* and the subsequently estimated Si – stock in this area. The clear spectral distinction between similar signatures prevents many other species from proper classification in general and in particular in the case of only few available spectral bands. This limitation can be

diminished by choosing an appropriate date for image acquisition, when the predominately green leaves of *C. epigejos* and *P. australis* enhance the optical contrast against the background reflectance of herbs, mosses and lichens. This cannot avoid the inclusion of other grass-like species such as *F. rubra* or *B. sylvaticum* in one of the *C. epigejos* or *P. australis* classes and the confusion with litter in case of sparse vegetation cover.

The majority of studies using the original NDVI, other VIs or combinations of VIs have been derived from satellite imagery

at landscape level (e. g. short grass prairie, Anderson et al., 1993; short grass steppe, Todd et al., 1998; rangeland, Mundava et al., 2014; different types of grassland and temperate steppe, Meng et al. 2018) showing poor up to moderate correlations between VIs and above ground biomass. R²s range between 0 and 0.6 for either total biomass or fractions of biomass caused by insufficient spatial and spectral resolution but mainly by background reflectance of soil, shadow or non-photosynthetic plant components such as standing dead plants or litter. The immense quantity of studies which evaluated different sensors

with numerous VIs at various scales and environments hamper a clear assessment of our results. While several studies reported no or minor improvements in the relationships between red edge VIs involving the wavelength region between 680 nm to 750 nm and vegetation parameters (e. g. Cui and Kerekes, 2018; Easterday et al., 2019), other studies carried out over heterogeneous forest stands have proven the red edge reflectance to be sensitive to chlorophyll content while largely unaffected by structural properties and crown shadow (e.g. Zarco-Tejada et al., 2018, Xu et al., 2019). This explains in parts the

outperformance of our results compared to the aforementioned studies and the results presented here, in particular for *P. australis*, which show a drastic decrease of R²s when NIR reflectance bands where used instead of the red edge band. The most important benefit of UAS-based remote sensing of phytogenic Si stocks is its potential to cover the heterogeneity in plant biomasses and thus phytogenic Si stocks. This is in contrast to previous studies, which assumed identical biomasses for Si stock quantifications (e.g., Cornelis et al., 2010; Sommer et al., 2013; Turpault et al., 2018). In addition, these studies mainly

focused on a single plant species in a given ecosystem. Contrary, UAS-based remote sensing enables to detect biomasses of



different plant species simultaneously, and thus to quantify Si stocks in a lifelike way. Furthermore, UAS-based remote sensing enables the detection of plant biomasses in larger areas, i.e., at a landscape scale, which is also in contrast to previous studies, which used results of small study plots for an extrapolation to larger spatial units up to a global scale implicitly assuming similar environmental conditions (e.g., Carey and Fulweiler, 2012). Due to the fact that aboveground biomass of plants seems

to be the main factor of corresponding phytogenic Si stock quantities (the variations in Si content in a plant species in a given ecosystem are considerably lower, especially when we assume that the phylogenetic position of a plant - rather than environmental factors - determines potential plant Si content, see, e.g., Hodson et al. 2005), a detection of biomass heterogeneities via UAS-based remote sensing represents a promising tool for the quantification of lifelike phytogenic Si pools at landscape scales.

## 5 Conclusions

As both species, *C. epigejos* as well as *P. australis*, analysed at Chicken Creek show a wide range of biomass (0.1 - 98 g m$^{-2}$), the use of mean biomasses in Si stock calculations or Si cycling models generally leads to deviations, thus substantial Si stock underestimations or overestimations. For a profound understanding of Si cycling in general and the influence of land use in particular, detailed information on the small-scale spatial distribution of plant related Si stocks based on an accurate biomass

assessment is urgently needed. This information will help us to understand the interaction between edaphic factors, plant distribution, ecosystem productivity (biomass), and anthropogenic desilicification on a local (i.e. site-specific) scale. Due to the fact that ultrahigh resolution imagery captured by UASs is capable of differentiating between Si accumulating species and respective fresh biomass and litter, this technique is also a promising tool for the detailed assessment of Si fluxes in grasslands. As Si content of litter have been recognized as an important driver of decomposition rates (Schaller et al., 2016, 2017),

information on litter Si stocks will further help us to better understand ecosystem biogeochemistry in general.

## Appendix A

### Appendix A1: UAS, camera specification, and mission settings

The Carolo P360 is a fixed wing construction, developed by the Institute of Aerospace Systems of the Technical University Braunschweig (Fig X). With a wingspan of 3.6 m and a take-off weight of almost 22.5 kg including the complete battery set

for the electric drive motor, the MINC autopilot system including servo actuators and the payload (sensors and control unit), the UAS is capable to carry an additional payload of approximately 2.5 kg. The battery set allows flight durations of approximately 40 min at ground speeds between 20 m s$^{-1}$ and 30 m s$^{-1}$ including the time for climbing and landing. The multispectral camera Mini-MCA 12 is a compact modular construction integrating two basic modules into one rugged chassis. Each module consists of an array of six individual CMOS sensors (1280 × 1024 pixels; pixel size 5.2 µm), lenses

(focal length 8.5 mm) and mountings for user definable band-pass filters. The filter configuration and specific properties are summarized in Table A1.





Mission settings followed the recommended cruising speed of 25 m s$^{-1}$ and the camera exposure time of 2 s resulted in an overlap in flight direction of approximately 50%. A distance of 40 m between the flight paths required to achieve a sufficient across flight overlap of at least 60%. Twenty-two waypoints were predefined, each marking a start- and endpoint of 10 parallel

flight paths with a total length of 8.6 km including the loop lines. In order to maintain the GSD of ~ 0.1 m, terrain effects were compensated by setting the flight altitude to 163 m for the northern- and 153 m for the southern waypoints. A total of 2556 individual (≙ 213 multispectral) images were captured during the mission.

**Table A1: Filter configuration of the Mini-MCA 12 and optical properties of the mounted filters. For band 2 (b) no fact sheet has been provided.**

| Band | Center Wavelength [nm] | FWHM* Coordinates (Bandwidth) [nm] | Bandwidth (10%) [nm] | Peak Transmission [%] |
|------|------------------------|-------------------------------------|----------------------|------------------------|
| 1 | 471 | 466.0 – 475.1  (9.1) | 12.8 | 68.3 |
| 2 | 515 | N/A (≈10.0) | N/A | N/A |
| 3 | 551 | 545.5 – 555.6 (10.1 ) | 14.8 | 56.4 |
| 4 | 613 | 607.7 – 617.8 (10.2 ) | 14.2 | 67.6 |
| 5 | 658 | 653.4 – 662.9  (9.5 ) | 13.6 | 69.2 |
| 6 | 713 | 708.1 – 717.7  (9.6 ) | 13.4 | 63.0 |
| 7 | 761 | 756.2 – 766.7 (10.5) | 14.7 | 71.9 |
| 8 | 802 | 797.3 – 807.3 (10.1) | 14.5 | 56.3 |
| 9 | 831 | 826.3 – 835.8  (9.5) | 13.1 | 55.3 |
| 10 | 861 | 856.4 – 866.4 (10.1) | 14.0 | 64.2 |
| 11 | 899 | 891.3 – 907.7 (16.4) | 22.9 | 63.6 |
| 12 | 953 | 933.0 – 973.8 (40.8) | 58.2 | 69.6 |

**Appendix A2: Image post-processing chain**

Post-processing of MCA imagery aims at the conversion of raw digital numbers (DN) into georeferenced at-surface reflectance images. This multistage procedure consists of three major components (i) radiometric image correction and (ii) transformation

of sensor coordinates into a geographic coordinate system and image alignment and (iii) absolute radiometric calibration. The radiometric image correction includes periodic and checkered pattern noise reduction, correction of sensor-based illumination fall-off (vignetting), horizontal band noise removal (caused by the progressive shutter of CMOS sensors) and lens distortion. The transformation of sensor coordinates includes the fusion of recorded GPS measurements with collected images, band-wise automated aerial triangulation (AAT), the minimizing of remaining geometric distortions and the alignment of single bands to

one multispectral image using ground control points (GCPs). Due to homogenous environmental conditions during the 10 minutes of image acquisition (weather and illumination geometry), we renounced the conversion of measured DNs into at-





surface reflectance, which is required for the retrieval of physical parameters of vegetation canopies or bare soil properties because a recorded DN is not only a function of the spectral characteristics of vegetation or soils but also of environmental condition (Moran et al., 1995). A detailed description of the multistage procedure is beyond the scope of this paper. Thus the

following paragraphs give a brief overview of the basic methods used in this study to generate one georeferenced multispectral image from the recorded raw images.

Radiometric corrections comprise noise reduction, correction for vignetting and lens distortion effects. The dark offset subtraction technique proposed by (Kelcey and Lucieer, 2012) reduces the noise component of an image by subtracting the average per-pixel noise calculated from 120 repetitions captured in a completely darkened environment for each of the 12

sensors. The method used for the correction of vignetting effects basically uses a look-up table (LUT) for each sensor, composed of correction factors for each pixel derived from flat field imagery (Mansouri et al., 2005). These were generated by capturing multiple images of an evenly illuminated white cardboard with almost lambertian properties and constant spectral characteristics over VIS and NIR wavelengths. In a first step the per-pixel average and the corresponding standard deviation were calculated from a total of 10 images for each of the 12 sensors at different exposure levels, followed by a subtraction of

the respective dark offset imagery.

To account for the horizontal band noise induced by the progressive shutter of the camera, a shutter correction factor has been calculated (Wehrhan et al., 2016).

Finally, a correction technique for lens distortions is applied. The plumb-line approach described in the Brown–Conrady model (Hugemann, 2010) is implemented in the PhotoScan-Pro V.1.7. software (Agisoft LLC, St. Petersburg, Russia). The model requires

the input of the focal length (8.5 mm) and the pixel size (5.2 μm). Internal and external orientation of each camera (band) is then estimated automatically from the geometry of an image sequence during the image alignment process (Dall'Asta and Roncella, 2014).

Mosaicking and geo-referencing using The PhotoScan-Pro workflow involves common photogrammetric procedures including the search for conjugate points by feature detection algorithms used in the bundle adjustment procedure, approximation of camera positions and orientation, geometric image correction, point cloud and mesh creation, automatic georeferencing and

finally the creation of an orthorectified mosaic (Conçalves and Henriques, 2015). This workflow was applied to each of the 10 bands independently. The ERDAS Imagine software (Hexagon Geospatial, Norcross, GA, U.S.) was then used to improve the spatial accuracy and to transform the single bands to the local coordinate system ETRS 89 UTM 33 by using the precisely measured raster point coordinates. Finally the 10 bands were stacked to a single multispectral image.

For more details regarding description of methods, used materials and technical equipment the reader is referred to (Wehrhan

et al., 2016).



## Appendix B

### Appendix B1: Image Classification

A supervised pixel-based classification of a natural ecosystem needs a clear understanding of the nature and the expected results. The result is governed (i) by the spectral and spatial resolution of input imagery, (ii) the biodiversity, (iii) the
morphology of the vegetation layer and (iiii) the selection of adequate training areas. The ultrahigh resolution easy allows the identification of larger objects such as trees, patches of bare soil and some individuals of *P. australis* but is still too low to identify individuals of *C. epigejos*. Due to almost unique spectral properties of green grass-like species, the number and the bandwidths of spectral MCA bands is insufficient for a clear distinction. The morphology of the vegetation layer determines whether a species is visible to the sensor or hidden by another species, e.g. trees or *P. australis* plants prevent the classification
of an underneath growing *C. epigejos* population. Finally the selection of training areas determines the classification quality. Selecting small training areas with little statistical variation in the signatures may result in large unclassified areas and vice versa. The separabilty between classes will be diminished if the signatures of the respective training areas are to some extent similar. As it is typical for a supervised maximum-likelihood classification, several trials are necessary to define an appropriate number of representative training areas with sufficient statistical separabilty of signatures. Taking all aforementioned aspects
into account, class definition is driven by the objectives addressed in this particular study. We are aware that these classes in parts do not coincide with vegetation or species communities as they are defined in the terminology of ecologist or biologists. However, for simplification purposes the term land cover class (LCC) was used.

### Author contribution

Conceptualization: M.S.; Formal analysis: M.W., D.P. and D.K.; Funding acquisition: M.S.; Investigation: M.W., D.P. and
D.K.; Methodology: M.W. and M.S.; Project administration: M.S.; Resources: M.S.; Validation: M.W., D.P, D.K. and M.S.; Visualization: M.W.; Writing - original draft preparation: M.W., D.P. and M.S.; Writing – review & editing: M.W., D.P. and M.S..

### Competing interests

The authors declare that they have no conflict of interest.

### Acknowledgements

D.P. was funded by the Deutsche Forschungsgemeinschaft (DFG) under grant PU 626/2-1 *(Biogenic Silicon in Agricultural Landscapes (BiSiAL) – Quantification, Qualitative Characterization, and Importance for Si Balances of Agricultural Biogeosystems)*.




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
