# Peer review of "Spatial patterns of aboveground phytogenic Si stocks in a grass-dominated catchment – Results from UAS based high resolution remote sensing"

_Biogeosciences, 2021_

## Author Comment (AC1)

I really appreciate Editor in-chief to invite me to review the manuscript by Marc Wehrhan et al. This is a very interesting MS, offering a fruitful experimental data and nice findings using Unmanned Aerial System (UAS). This is novelty and originality. Indeed, the abstract should emphasize new findings and their significance, with some experimental results/data.

Thank you very much for your kind words and the thorough revision of our manuscript. In the revised version of our manuscript we will adjust the abstract according to your comment.

In Introduction section, authors should give an objective summary to give a promising gap regarding phytogenic Si and soil properties affecting silicon mobility, which is quite important in the new findings and their significance of this MS. Please check all relative recent references. In addition, here I am not English speaker, but still find some grammatical errors, so that it will be better to improve its English for a better understanding before further publication. Please see the below soem specific comments:

The English of our revised manuscript will be double-checked before resubmission. Furthermore, we will rework the introduction based on your insightful comments (please see our detailed answers below).

L. 15, this sentence should be rephrased since it is not clear to me 'most of these studies…. condition'

The sentence is rephrased and separated into three:
Most studies are deliberately designed on the plot scale to ensure low heterogeneity in soils and plant composition, hence similar environmental conditions. Due to the immanent spatial soil variability the transferability of results to larger areas, such as catchments, is therefore limited.

L.25, referring 'i.e., i.e., comparable to or markedly exceeding reported values for the Si storage in aboveground vegetation of various terrestrial ecosystems.' prefer authors to give experimental or analytical values/data'

In this case we would like to refrain from the presentation of detailed values/data and respective citations because it's not only one study we refer on. From our point of view this would be too much for an abstract and finally all studies will be presented in the discussion.

L.25, add ','after 'from our results…'

We will add a comma.

L.50, here, prefer to author should also refer that 'since soil properties affect soil silicon bioavailability, leading to the change in plant silicon content (see., Li et al., 2019., Plant and Soil 438 (1), 187-203 and others). In fact, any change in soil properties would largely affect silicon mobility and its accumulation in plants. It has been highlighted by recent studies, offering some nice evidences on this MS.

Okay, we will revise this sentence and also add some (recent) literature to underline the importance of soil properties for Si bioavailability, and thus for Si uptake by plants.

Line 37-38: Please cite relevant references to support 'in most terrestrial ecosystems phytogenic Si…' (e.g., Alexandre et al. 1997. Geochimica et Cosmochimica Acta 61, 677-682; Blecker et al. 2006., Global Biogeochemical Cycles, 20; Cornelis et al. 2010. Biogeochemistry, 97, 231-245.Yang et al. 2020., Geoderma, 361: 114036). In particular, once being returned into soil, this phytogenic Si is largely are competitive with pedogenic silica, boosting the biological recycling of Si (Li et al., 2020., Geoderma, 368, p.114308).

We will add references for our statement and also highlight the importance of recycling of phytogenic Si for Si uptake by plants.

L41-42: Other recent studies also reported that the grasses of the family Poaceae are generally Si accumulators.

We will add a more recent reference to support this point.

Line 72-78: is it important or necessary for this MS to introduce these studies?

We intended to emphasize the importance of UAV remote sensing for hardly accessible or protected ecosystems, where ultrahigh resolution is required but cannot be provided by satellite imagery. We will rephrase and shorten the paragraph as follows to focus on this intention.

"The recent development of Unmanned Aerial Systems (UAS) offers new options for ultrahigh-resolution observations at landscape and catchment scale. Numerous missions have been conducted over hardly accessible areas or protected ecosystems such as wetlands (Strecha et al., 2012; Zweig et al., 2015), riparian zones of lakes and rivers (Husson et al., 2014; Husson et al., 2016), estuarine tidal flats (Kaneko and Nohara, 2014), riparian forests (Dunford et al., 2009) and Antarctic moss beds (Turner et al., 2014). Most of the studies delineated the patchy and small-scale distribution of plant communities and identified individual species by using of-the-shelf (partly modified) compact digital cameras or narrow-band multispectral sensors providing an adequate sub-decimeter resolution in VIS and NIR spectral wavelengths.

Successful preprocessing workflows were developed for UAS imagery as a prerequisite for accurate image interpretation (Laliberte et al., 2011; Berni et al., 2009; Kelcey and Lucieer, 2012; Lelong et al., 2008; Wehrhan et al., 2016)."

Line 123-124 and Line 127-128: When the aboveground biomass of *C. epigejos* and *P. australis* were sampled? Is it in 2014? Please specify.

Aboveground biomass have been sampled a couple of days after image acquisition. We will add this information.

L256-257, L265, L272, L320-321, L 324-325, L334-335, L355-356, 379-380, Line: Use italics when showing the name of the species. Please check throughout the manuscript.

We will correct this mistake throughout the MS.

L324 (Figure 6): Please change the title of *y*-axes to "fresh biomass (green shoot)" in Figure 6a, and change the title of *y*-axes to "fresh biomass (green shoot + litter)" in Figure 6b.

We will change titles of y-axes in both figures.

[Figure]

Li334 (Figure 7): Please change the title of *y*-axes to "dry biomass (green shoot)" in Figure 7a, and change the title of *y*-axes to "dry biomass (green shoot + litter)" in Figure 7b.

We will change titles of y-axes in both figures and, if you agree, also the respective x-axes.

[Figure]

L372-403: a bit confusing about this section. Right now, the relationship between Si stocks of *C. epigejos*, *P. australis* and site properties was dubious just by comparing the variation trends between Si stocks and examined soil properties in different zones (e.g., Line 388-389: Among the examined soil properties, means of clay content (Fig. 10a) show a corresponding trend with respect to Si accumulation in dry biomass of *C. epigejos* for all three zones.). Could you perform statistical analyses between Si stocks (*C. epigejos*, and *P. australis*, respectively) and different site properties to show their relationship? At least Pearson correlation analysis is needed.

We agree that a correlation analyses could underline our conclusions in general, and thus will add corresponding results to the revised version of our manuscript.

Line 379 (Figure 9) and Line 391 (Figure 10): What does the data on the top of box represent? Mean or median? What does the bottom and top bars represent? Please specify.

We will amend the descriptions of figures 9 and 10 to explain (inclusively added letters a and b; indicating statistical significance of differences between zones according to Kruskal-Wallis ANOVA (p < 0.05)).

"Figure 9: Data distribution (mean, upper and lower quartile, minimum and maximum) of Si stocks of *C. epigejos* (a) and *P. australis* (b) at grid points in the western, eastern and southern zone. Numbers represent zonal means. Letters (a and b) indicate statistical significance of differences between zones according to Kruskal-Wallis ANOVA (p < 0.05). Note the different scaling of the y-axes."

"Figure 10: Data distribution (mean, upper and lower quartile, minimum and maximum) of soil properties and nutrients at grid points in the western, eastern and southern zone. Figures depict the distribution of clay content (a), Tiron extractable amorphous Si (SiTiron) (b), water soluble Si (Si – $H_2O$) (c), nitrogen (d), potassium (e), and phosphorus (f). Numbers represent zonal means. Letters (a and b) indicate statistical significance of differences between zones according to Kruskal-Wallis ANOVA (p < 0.05)."

Line 379 (Figure 9) and Line 391 (Figure 10): Right now, the readers do not know whether there are significant differences between zones. Could you perform significance test between the zones to show the significant differences?

Please refer to the answer above. We will add a sentence at the end of the paragraph (Line 387)

"According to Kruskal-Wallis ANOVA (p < 0.05), differences in mean Si stocks are statistically significant between the western and eastern zone for *C. epigejos* and between the southern and the other two zones for *P. australis*."

We will extend the sentence in Line 389 as follows:

"Among the examined soil properties, means of clay content (Fig. 10a) show a corresponding trend with respect to Si accumulation in dry biomass of *C. epigejos* for all three zones including the significant difference between the western and eastern zone."

Line 401: Could you offer the data of soil moisture to support this conclusion: "As stated before, the occurrence of *P. australis* is governed by soil moisture conditions".

This is more a general statement based on the scientific knowledge about Phragmites water demands and occurences in landscapes. At the "Chicken Creek catchment" only four sites outside the groundwater influenced areas were equipped with devices for continuous soil moisture monitoring. Groundwater gauges were distributed over the entire catchment. We will add information about these data and the hydrology of the catchment in the revised version, e.g. from Hölzel et al. JHydrol., 2013.

Line 432-434: also recommend some latest literatures (straw remove, return, land use and management change) to support this point. e.g., "Li and Delvaux 2019. *GCB Bioenergy* 11, 1264–1283" and "Yang et al. 2020. *Plant and Soil*, 454:343–358".

Thanks, we will add these important and more recent studies here.

L442-443: I confusion whether the climatic factors could govern the composition and structure of plant communities at Chicken Creek. I think the differences of climatic conditions may be negligible at such small catchment.

Here, we refer to the general influence of climatic conditions on the composition and structure of plant communities and not on the spatial variability within the fenced area. We agree, the climatic conditions on catchment scale are less than negligible. We propose to rephrase the paragraph as follows:

"In general, the composition and structure of plant communities and the spatial distribution of plant biomass is mainly controlled by climatic, edaphic (e.g., soil moisture/texture, pH, and nutrients) and geographic-historic factors as well as by species interactions (e.g., consumption by herbivores) and (anthropogenic) perturbations (e.g., Polis 1999). At Chicken Creek consumption of plants by herbivores can be generally excluded, because the total study area is fenced. Studies of Zaplata et al. (2011a, 2013) indicated that differences in vegetation dynamics at Chicken Creek can be directly derived from slight differences in edaphic conditions resulting from construction work with large machines (Gerwin et al., 2010)."

L506-515: In my side, the current Conclusion is more like Discussion or Outlook. Prefer to move this paragraph to the end of Discussion section.

Good point, thanks for this suggestion! We will move this paragraph.

L505: recommend the authors reconsider the Conclusions section by combining the main findings and significance of this study or answering the three major research questions raised in Introduction section.

Thanks again. We will rework the conclusions in our revised manuscript to better illustrate the significance of our results.

---

## Author Comment (AC2)

First of all, we would like to thank you for your comments on our manuscript. Please find our corresponding answers below.

Line 135, the authors should confirm that both "4.000 rpm" and "10.000 rpm" are corrected. Based on my experience, they should be "4000 rpm" and "10000 rpm" in this context.

You are right. We will correct this in our revised manuscript.

Line 144, "Soil samples were analysed on various physicochemical soil properties". The second "soil" should be deleted.

Correct, we will delete "soil"

Figure 3, the numbered signatures were confused. Why the small numbers (i.e., 01-07) were shown in Figure 3b while the big numbers (i.e., 08-16) present in Figure 3a?

You are right, the most important signatures should be presented first. We will switch the figures.

Table 2. Is it more suitable to show the data as box chart rather than table?

Good point. We will try out a diagram for these data and choose the one (table or diagram), which is more intuitively accessible for the readers.

Table 3. The unit of RMSE should be corrected for C. epigejos, P. australis, and P.australis (litter incl. brown shoot). Specifically, "-2" should be labeled as superscript.

We overlooked that and will correct it

Figure 6. Why the R2 is lost for C. epigejos in Figure 6b?

We will reinsert the $R^2$

Figure 9. Are there any statistical differences among West, East, and South for both C.epigejos and P. australis?

According to Kruskal-Wallis ANOVA ($p < 0.05$) significant differences exist between the western and the eastern zone for *C. epigejos* and between the southern and the other two zones for *P. australis* indicated by different letters (a and b; see modified figures below).

[Figure]

We will amend the description of Figure 9 to explain added letters: "Distribution (mean, upper and lower quartile, minimum and maximum) of Si stocks of *C. epigejos* (a) and *P. australis* (b) at grid points in the western, eastern and southern zone. Numbers represent zonal means. Letters (a and b) indicate statistical significance of differences between zones according to Kruskal-Wallis ANOVA (p < 0.05). Note the different scaling of the y-axes.

We will add a sentence at the end of the paragraph (Line 387) "According to Kruskal-Wallis ANOVA (p < 0.05), differences in mean Si stocks are statistically significant between the western and eastern zone for *C. epigejos* and between the southern and the other two zones for *P. australis*."

---

## Author Response (AR1)

**Reply to reviewer 1**

Again, we would like to thank reviewer 1 for the comments on our manuscript. Please find our corresponding answers below.

In Introduction section, authors should give an objective summary to give a promising gap regarding phytogenic Si and soil properties affecting silicon mobility, which is quite important in the new findings and their significance of this MS. Please check all relative recent references. In addition, here I am not English speaker, but still find some grammatical errors, so that it will be better to improve its English for a better understanding before further publication. Please see the below soem specific comments:

The English of our revised manuscript was double-checked before resubmission. Furthermore, we reworked the introduction based on your insightful comments (please see our detailed answers below).

L. 15, this sentence should be rephrased since it is not clear to me 'most of these studies…. condition'

We rephrased the sentence and separated it into two:
Most studies are deliberately designed on the plot scale to ensure low heterogeneity in soils and plant composition, hence similar environmental conditions. Due to the immanent spatial soil variability, the transferability of results to larger areas, such as catchments, is therefore limited.

L.25, referring 'i.e., i.e., comparable to or markedly exceeding reported values for the Si storage in aboveground vegetation of various terrestrial ecosystems.' prefer authors to give experimental or analytical values/data'

We assume that you can follow our argument as stated in the first reply.

L.25, add ','after 'from our results…'

We added a comma here.

L.50, here, prefer to author should also refer that 'since soil properties affect soil silicon bioavailability, leading to the change in plant silicon content (see., Li et al., 2019., Plant and Soil 438 (1), 187-203 and others). In fact, any change in soil properties would largely affect silicon mobility and its accumulation in plants. It has been highlighted by recent studies, offering some nice evidences on this MS.

We added some (recent) literature to underline the importance of soil properties for Si bioavailability, and thus for Si uptake by plants and corresponding consequences for plant performance.

Line 37-38: Please cite relevant references to support 'in most terrestrial ecosystems phytogenic Si…' (e.g., Alexandre et al. 1997. Geochimica et Cosmochimica Acta 61, 677-682;

Blecker et al. 2006., Global Biogeochemical Cycles, 20; Cornelis et al. 2010. Biogeochemistry, 97, 231-245.Yang et al. 2020., Geoderma, 361: 114036). In particular, once being returned into soil, this phytogenic Si is largely are competitive with pedogenic silica, boosting the biological recycling of Si (Li et al., 2020., Geoderma, 368, p.114308).

We added references for our statement and also highlight the importance of recycling of phytogenic Si for Si uptake by plants.

L41-42: Other recent studies also reported that the grasses of the family Poaceae are generally Si accumulators.

We added a more recent reference to support this point.

Line 72-78: is it important or necessary for this MS to introduce these studies?

We rephrased and shortened the paragraph according your justified question.

Line 123-124 and Line 127-128: When the aboveground biomass of *C. epigejos* and *P. australis* were sampled? Is it in 2014? Please specify.

Aboveground biomass have been sampled a couple of days after image acquisition. We added this information to the beginning of the paragraph.

L256-257, L265, L272, L320-321, L 324-325, L334-335, L355-356, 379-380, Line: Use italics when showing the name of the species. Please check throughout the manuscript.

We corrected this mistake throughout the manuscript.

L324 (Figure 6): Please change the title of *y*-axes to "fresh biomass (green shoot)" in Figure 6a, and change the title of *y*-axes to "fresh biomass (green shoot + litter)" in Figure 6b.

Titles of y-axes were changed according your recommendation in Fig. 6.

Li334 (Figure 7): Please change the title of *y*-axes to "dry biomass (green shoot)" in Figure 7a, and change the title of *y*-axes to "dry biomass (green shoot + litter)" in Figure 7b.

Titles of y-axes were changed according your recommendation and, inspired by your proposal, we changed the titles of x-axes.

L372-403: a bit confusing about this section. Right now, the relationship between Si stocks of *C. epigejos*, *P. australis* and site properties was dubious just by comparing the variation trends between Si stocks and examined soil properties in different zones (e.g., Line 388-389: Among the examined soil properties, means of clay content (Fig. 10a) show a corresponding trend with respect to Si accumulation in dry biomass of *C. epigejos* for all three zones.). Could you perform statistical analyses between Si stocks (*C. epigejos*, and *P. australis*, respectively) and different site properties to show their relationship? At least Pearson correlation analysis is needed.

We agree and added results of correlation analyses to the revised version of our manuscript.

Line 379 (Figure 9) and Line 391 (Figure 10): What does the data on the top of box represent? Mean or median? What does the bottom and top bars represent? Please specify.

We changed the captions of figures 9 and 10 to explain the respective features.

Line 379 (Figure 9) and Line 391 (Figure 10): Right now, the readers do not know whether there are significant differences between zones. Could you perform significance test between the zones to show the significant differences?

We added letters (a and b; see modified figures below) indicating significant differences according to Kruskal-Wallis ANOVA (p < 0.05), changed the caption of figures 9 and figure 10. We amended respective sentences at the end of the paragraph starting at Line 387.

Line 401: Could you offer the data of soil moisture to support this conclusion: "As stated before, the occurrence of *P. australis* is governed by soil moisture conditions".

We assume that you accepted our explanation for this general statement and remained this conclusion unchanged.

Line 432-434: also recommend some latest literatures (straw remove, return, land use and management change) to support this point. e.g., "Li and Delvaux 2019. *GCB Bioenergy* 11, 1264–1283" and "Yang et al. 2020. *Plant and Soil*, 454:343–358".

Thanks, we added these important and more recent studies here.

L442-443: I confusion whether the climatic factors could govern the composition and structure of plant communities at Chicken Creek. I think the differences of climatic conditions may be negligible at such small catchment.

We rephrased the whole paragraph in order to clarify this point (please refer to the first reply)

L506-515: In my side, the current Conclusion is more like Discussion or Outlook. Prefer to move this paragraph to the end of Discussion section.

Thanks for this suggestion again. We followed your argumentation and moved the paragraph to the end of the discussion section.

L505: recommend the authors reconsider the Conclusions section by combining the main findings and significance of this study or answering the three major research questions raised in Introduction section.

Thanks again. We reworked the conclusions in our revised manuscript to better illustrate the significance of our results.

**Reply to reviewer 2**

Again, we would like to thank reviewer 2 for the comments on our manuscript. Please find our corresponding answers below.

Line 135, the authors should confirm that both "4.000 rpm" and "10.000 rpm" are corrected. Based on my experience, they should be "4000 rpm" and "10000 rpm" in this context.

We corrected this mistake.

Line 144, "Soil samples were analysed on various physicochemical soil properties". The second "soil" should be deleted.

We deleted the second "soil", which is, of course, an unneeded repetition.

Figure 3, the numbered signatures were confused. Why the small numbers (i.e., 01-07) were shown in Figure 3b while the big numbers (i.e., 08-16) present in Figure 3a?

We switched the figures according the logical order of class numbers.

Table 2. Is it more suitable to show the data as box chart rather than table?

We tried out a diagram for these data, but kept the table, because it appeared more intuitively accessible for the readers to our mind.

Table 3. The unit of RMSE should be corrected for C. epigejos, P. australis, and P.australis (litter incl. brown shoot). Specifically, "-2" should be labeled as superscript.

We corrected this mistake.

Figure 6. Why the R2 is lost for C. epigejos in Figure 6b?

Figure 6 was corrected according your hint.

Figure 9. Are there any statistical differences among West, East, and South for both C.epigejos and P. australis?

We performed a statistical analysis, added letters (a and b; see modified figures below) indicating significant differences according to Kruskal-Wallis ANOVA ($p < 0.05$), changed the caption of figure 9 and amended a respective sentence at the end of the paragraph (Line 387).